# Cancer detection for small-size and ambiguous tumors based on semantic FPN and transformer

**Jingzhen He¹, Jing Wang ²\*, Zeyu Han³, Baojun Li⁴, Mei Lv⁵, Yunfeng Shi²\***

**1** Department of Radiology, Qilu Hospital of Shandong University, Jinan, China, **2** School of Information Science and Engineering, Shandong Normal University, Jinan, China, **3** School of Mathematics and Statistics, Shandong University, WeiHai, China, **4** College of Vocational Education, Dezhou University, Dezhou, China, **5** School of Physical Education Department, Shandong Women's University, Jinan, China

\* jingwang1551@163.com (JW); yunfeng@sdnu.edu.cn (YS)

**Data Availability Statement:** All data files are published and available from the CBIS-DDSM database. (http://www.eng.usf.edu/cvprg/Mammography/Database.html).

## Abstract

Early detection of tumors has great significance for formative detection and determination of treatment plans. However, cancer detection remains a challenging task due to the interference of diseased tissue, the diversity of mass scales, and the ambiguity of tumor boundaries. It is difficult to extract the features of small-sized tumors and tumor boundaries, so semantic information of high-level feature maps is needed to enrich the regional features and local attention features of tumors. To solve the problems of small tumor objects and lack of contextual features, this paper proposes a novel Semantic Pyramid Network with a Transformer Self-attention, named SPN-TS, for tumor detection. Specifically, the paper first designs a new Feature Pyramid Network in the feature extraction stage. It changes the traditional cross-layer connection scheme and focuses on enriching the features of small-sized tumor regions. Then, we introduce the transformer attention mechanism into the framework to learn the local feature of tumor boundaries. Extensive experimental evaluations were performed on the publicly available CBIS-DDSM dataset, which is a Curated Breast Imaging Subset of the Digital Database for Screening Mammography. The proposed method achieved better performance in these models, achieving 93.26% sensitivity, 95.26% specificity, 96.78% accuracy, and 87.27% Matthews Correlation Coefficient (MCC) value, respectively. The method can achieve the best detection performance by effectively solving the difficulties of small objects and boundaries ambiguity. The algorithm can further promote the detection of other diseases in the future, and also provide algorithmic references for the general object detection field.

## Introduction

Breast cancer is one of the most deadly malignancies among women worldwide [1]. The difficulty of accurately screening for early-stage tumors has increased mortality from breast cancer [2, 3]. Therefore, early identification of breast cancer is necessary for proper treatment and

**Funding:** We provide repository information for our data at acceptance. All data files are published and available from the CBIS-DDSM database. "http://www.eng.usf.edu/cvprg/Mammography/Database.html".

recovery. With the increasing quantity of mammograms in hospitals, manual reading has become complex and time-consuming for radiologists. Computer-aided detection system assists radiologists in diagnosis, with the goal of reducing screening time and improving early detection accuracy [4–6]. However, there are some challenges in image feature extraction and accurate detection of early breast mammograms. Firstly, there are some differences between cancerous and noncancerous breast tissue on imaging, but the difference in early cancer diagnosis is minimal. The low signal-to-noise ratio (SNR) [7, 8] of mass compared with the surrounding tissue leads to feature extraction difficulties in the lesion region. Secondly, the varying size of cancer masses is one of the challenges of detection, especially when small ones are difficult to detect. The third challenge is the blurring of tumor boundaries, which may cause visual confusion leading to inefficient target regression. Therefore, cancer detection remains a challenging task.

In small object detection, early research focused on feature extraction from small-sized regions. The classical Feature Pyramid Network (FPN) [9] algorithm achieves the extraction of multi-scale features through a top-down multi-level architecture. However, the upsampling operation used by FPN loses the position information of small objects. Recently, the popular transformer attention model [10, 11] can effectively capture objects regions based on encoder-decoder and attention mechanisms. In addition, Wu *et al.* [12] disentangle the sibling head into two independent branches for classification and localization. The feature coupling method can effectively improve the performance of small-size detection. Two characteristics of early medical images are small tumor areas and blurred borders. Thus, it is necessary to design an effective method to enrich high-resolution and local attention features with semantic information from multi-level feature maps.

To solve mentioned above problems, the paper proposes a detection framework for medical imaging, named Semantic Pyramid Network with a Transformer Self-attention (SPN-TS). As shown in Fig 1(A), without enough semantic information, low-level feature maps are difficult to detect small tumor objects and have good performance. Fig 1(A) shows the traditional FPN by top-down and horizontal connection. The way does not yield sufficient semantic information, resulting in poor feature extraction from small tumor regions. In this paper, we creatively design a new FPN feature extraction scheme, which changes the traditional connectivity and improves the semantic information. As shown in Fig 1(B), the

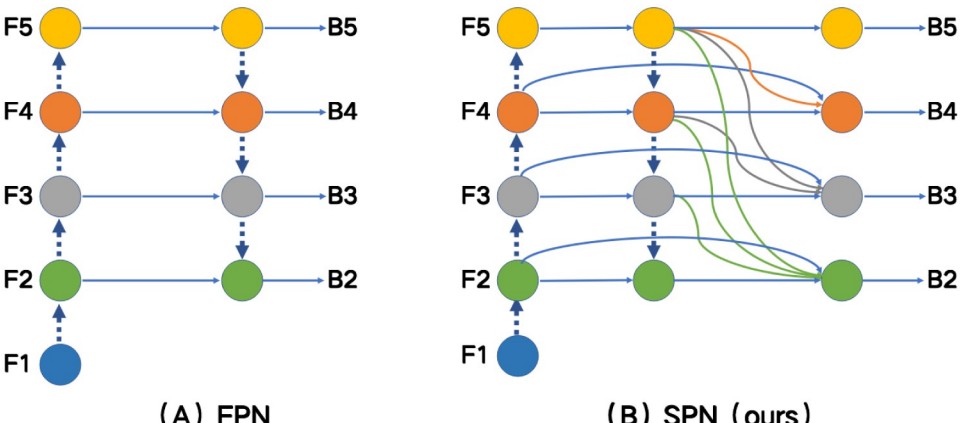

**Fig 1. The figure compares the original FPN with our proposed SPN connection.** *Fi* is multi-level features from layer 1 to 5, and $B_i$ is the output of the feature pyramid of the *i* level. (A) The traditional FPN is by top-down and horizontal connection. (B) Enriched the semantic features through the three steps of lateral connectivity, multiple up-sampling, and feature fusion.

SPN module enriches the semantic features through the three steps of lateral connectivity, multiple up-sampling, and feature fusion. In addition, we introduce the transformer attention mechanism for comprehensive feature learning in tumor image detection. The transformer mechanism abandons the traditional CNN. It can capture the effective regions of objects in the image during tumor detection. Finally, we conducted extensive experimental verification on the CBIS-DDSM, which is the Curated Breast Imaging Subset of the Digital Database for Screening Mammography. The contribution of this paper is mainly in the following three aspects.

1. The paper proposes an effective network called SPN-TS for cancer detection, containing semantic FPN and transformer self-attention mechanisms. It addresses the extraction of small tumor objects and the lack of contextual information.

2. The novel transformer attention mechanism is integrated into the extraction network. The network focuses on tumor region features by attention mechanism and location encoding.

3. The paper also decouples the classification and regression branches to improve the classification confidence. The experiment conducted an extensive evaluation of the CBIS-DDSM dataset to illustrate the effectiveness of the SPN-TS method in detecting small objects.

The rest of this paper is organized as follows. The Related Work summarizes the cancer diagnosis and object detection. In Materials and methods section, it is described in detail for addressing problems. In section Results, experimental design and results analysis are presented to compare with the existing studies on breast cancer detection and object detection, respectively. Discussion section mainly describes the discussion of breast disease. Finally, the last includes the conclusions of the work and future work.

## Related work

Mammography is currently an effective method to detect breast cancer and its formative stage for breast cancer screening [13, 14]. In earlier studies, the Support Vector Machines (SVM) algorithm has a good classification performance and is mostly used for breast tumor classification and identification tasks [15–18]. V. Jitendra *et al.* [19] proposed a system based on the Histogram of Orientation Gradients (HOG) descriptor to classify objects using SVM. C. Muramatsu *et al.* [18] use texture features to classify breast lesions into benign and malignant categories. In reference [20], multi-scale region growth and wavelet decomposition were combined to locate a region of interest and achieved 96.19% detection accuracy of pathology images of breast cancer. In recent years, convolutional neural networks (CNN) and artificial intelligence technology have been widely used in object detection [21, 22]. The deep learning-based methods also were widely used for medical imaging [23–25]. Yang *et al.* [23] proposed a multi-perspective detection framework, which combines the convolution neural network of two views of mammogram image to predict the case of mammogram image classification. R. Platania *et al.* [26] proposed a YouOnlyLookOnce (YOLO) based CAD system called BC-DROID.

Object detection algorithms are generally applied to the automatic detection of cancer images, especially breast masses [27, 28]. Object detection algorithms are divided into two-stage and one-stage algorithms depending on the processing flow. Faster-RCNN, as a two-stage representative algorithm, generates region proposals through the Region Proposal Network (RPN) module, and then performs fine-grained classification and regression [9, 29–31]. The typical one-stage YOLO [26, 32] and SSD [33] detectors are designed to detect the

diversity of object sizes. In feature extraction, FPN [9] first builds a top-down architecture to extract features from multiple layers by connecting them laterally. In addition, Wu *et al.* [12] disentangle the sibling head into two independent branches for classification and localization. The DETR method [34] successfully integrates the Transformer self-attention mechanism into an object detection framework for detecting the central building blocks of the pipeline.

## Materials and methods

### Overview

This paper novelty proposes the Semantic Feature Pyramid network with Transformer Self-attention module, named SPN-TS, for tumor detection. The overall process is as follows. Firstly, this paper creatively designs a semantic FPN feature extraction scheme, which changes the traditional cross-layer connectivity way. Then, we introduce the transformer attention mechanism abandoning the traditional CNN method. The mechanism is used for comprehensive feature learning and boundaries feature extraction in tumor image detection. The overall architecture of SPN-TS method is shown in Fig 2. It is mainly divided into four parts, transformer attention module, semantic feature pyramid network, region proposal network, prediction of classification and regression module. Specifically, the transformer attention module stitches multiple attention features using the multi-headed self-attention mechanism and positional encoding. It can effectively capture the key regions and richer semantic features via the combined contextual information. Then, the semantic FPN network is creatively proposed to change the traditional cross-layer connectivity scheme. It is mainly used to enrich the semantic information of low-level feature maps and improve the

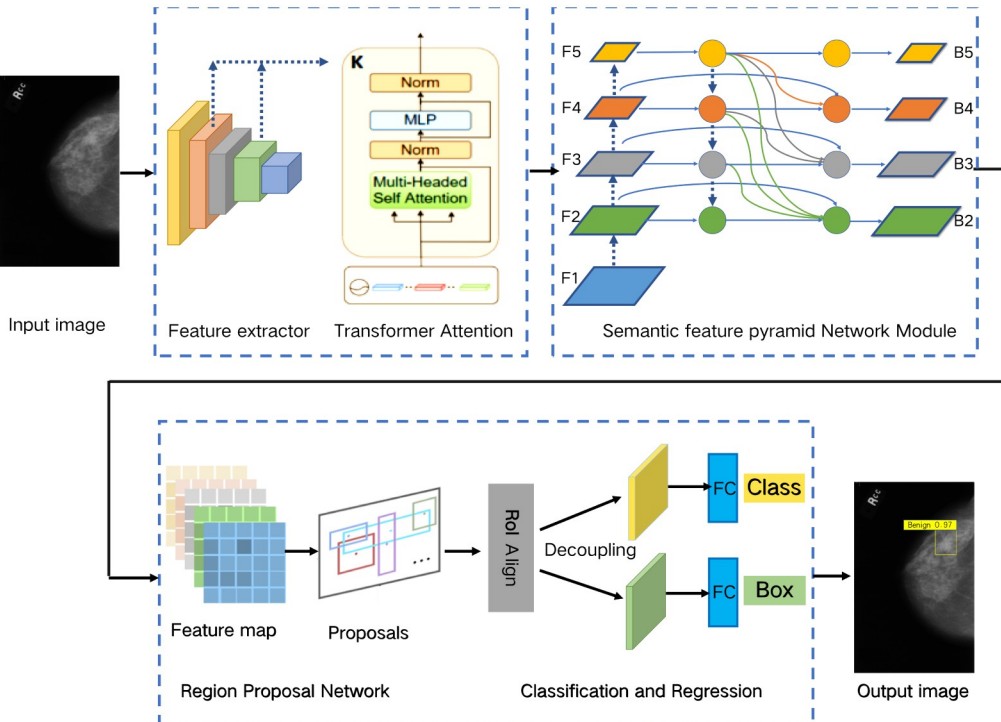

**Fig 2. The overview architecture of proposed SPN-TS, including the following four parts, transformer attention feature extractor, semantic feature pyramid network module, region proposal network and classification and regression module.**

detection performance of small tumor objects. In addition, the paper decouples the features of classification and regression branches and uses fully connected layers and convolutional layers for prediction, respectively.

## Semantic feature pyramid network

The paper proposes the semantic FPN network to enrich the semantic features by integrating high-level features into low-level feature maps and improve the detection performance of small tumor objects. As shown in Fig 2, the rich semantic information is obtained through the three steps of lateral connectivity, multiple up-sampling, and feature fusion. The newly connected feature maps are input into Region Proposal Network (RPN) and Region of Interest Alignment (ROIAlign) respectively. Then, the output results of ROIAlign are detected and regressed to fully integrate the spatial location and semantic information of the object, so that the detection model can better classify and localize the input images.

The first operation is lateral connection. We propose innovative lateral connections that preserve the resolution and semantic information of the current layer while facilitating the feature fusion operations in the subsequent steps. In Fig 1, $F_i$ are multi-level features from layer 1 to 5, and is the feature map of $I_{th}$ stage in the intermediate process through the transverse connection defined as $M_i$. It is also the corresponding output feature of the original FPN without feature fusion. At the same time, lateral connections are calculated by setting a 3 convolutional layer on each of the merged feature map to reduce the aliasing effect of up-sampling and integration.

Then, the second operation is lateral connection multiple up-sampling. To integrate multi-level features and preserve semantic information, we need to up-sampling feature maps $M_i$ to the corresponding size. The $M_2$ was obtained by fusion of three times down-sampling of $F_2$-$F_5$ at different scales. In short, the fusion operation in the cross-scale and dense pathway of SPN is described as:

$$\begin{cases} B_5 = F_5 \\ B_4 = f_4(F_4,\ Sp(F_4),\ up(F_5)) \\ B_3 = f_3(F_3,\ Sp(F_3),\ up(F_5),\ up(F_4)) \\ B_2 = f_2(F_2,\ Sp(F_2),\ up(F_5),\ up(F_4),\ up(F_3)) \end{cases} \tag{1}$$

where $f_i$ is the function of $i_{th}$ multi-scale feature fusion operation, $i$ is the layer. The high-level feature $B_{i+1}$ are propagated through the classical nearest interpolation function $up(.)$ for up-sampling. The $Sp$ is represented as skip connection.

In the feature fusion step, we integrate feature maps from different layers and different sizes, where $B_i$ is the output of the feature pyramid of $i$ level. For simplicity, we take the third layer as an example, and first scaled the high-level feature map ($F_4$) by a scaling factor of 2 using the nearest neighbor up-sampling method. Then, it is merged with the shallow feature maps $F_3$ and $F_2$, respectively, and then, we obtain $B$ with a 3 × 3 convolution layer. Finally, this feature map is fused into $B_2$ through a 3 × 3 convolutional layer, and the feature maps of other levels are obtained in the same way in turn. By completing the above steps, we will achieve an effective fusion of features from low to high.

## Transformer attention module

As a result of these improvements, the detection of small objects is not as good as expected. These approaches lack adjustments for small object and boundary, which is one of the reasons for the unsatisfactory performance of small object detection. Therefore, the paper integrated the transformer attention module into a network that contains a self-attention mechanism based on deconvolution and a skip connection. The structure of a complete transformer attention model is shown in Fig 3(a). The transformer attention module helps to extract clearer and richer semantic features, especially adge features, through combining contextual information.

As shown in Fig 3(b), the module applies $3 \times 3$ convolution in ResNet-50 to get the output results. Firstly, a new feature map is obtained by $1 \times 1$ convolution and $3 \times 3$ convolution of the input feature map. Then, it is added to the feature map obtained through transformer attention, followed by $1 \times 1$ convolution of the network and finally summed with the input feature map. The feature map obtained from the previous steps is the input for the next stage of FPN.

As shown in Fig 3(c), our improved transformer is connected across layers using normalization instead of regularization on the backbone network. The transformer layers are a stacked transformer encoders that extract the degradation features from the reconstructed data, with two sub-layers: Multi-Head Self-Attention and Feed-Forward (FFN). A transformer attention is composed of K stacks with identical layers. Each stack is divided into two sub-layers, a novel multi-headed self attention mechanism and a MLP feed-forward network. The residual connectivity operation common in ResNet, is used to feature normalization around each sublayer. In Fig 3, the first layer of multi-headed attention is the integration of multiple self-attention structures. The sub-layer designed to capture the dependencies between features and ignores their distances. Given the representation of the $(l-1)$ layer, $H^{(l-1)}$ and h parallel attention functions, the $i_{th}$ attention is defined:

$$head_i = Attention(H^{l-1} \cdot W_Q^l, \ H^{l-1} \cdot W_K^l, \ H^{l-1} \cdot W_V^l) \tag{2}$$

where $W_Q$, $W_K$, $W_V$ are projection weights. Let Q, K, and V denote query, key, and value, the

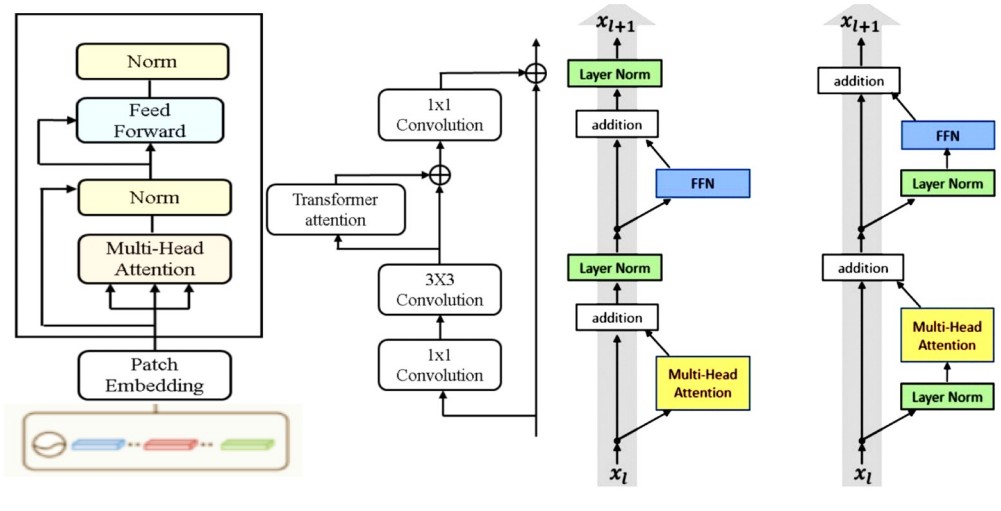

(a) Overall Architecture    (b) Operating details    (c) Contrast abstract representation

**Fig 3. The residual connectivity operation and attention module.** (a) Overall Architecture. (b) Operating details. (c) Contrast abstract representation.

scaled dot-product attention defined as follows,

$$Attention(Q, \ K, \ V) = softmax(\frac{Q \cdot K^T}{\sqrt{d_h}}) \cdot V \tag{3}$$

where $d_h = \frac{d}{h}$, which avoids extremely small gradients and produces a softer attention distribution. Then, the Multi-Head Attention is defined as follows,

$$Multi - Head(H^{l-1}) = [head_1, \ head_2, \ \dots \ , \ head_h] \cdot W^0 \tag{4}$$

where $W^0$ is a trainable weight.

Feed Forward has two different mappings, both linear and ReLU nonlinear, and is applied identically to each time step. Then, the method abtains $H^l$ from the previous multi-head ($H^{l-1}$) as follows,

$$H^l = FFN(Multi - Head(H^{l-1})) \tag{5}$$

$$FFN(x) = ReLU(x \cdot W_1 + b_1) \cdot W_2 + b_2 \tag{6}$$

The experimental data is fed into the self-attentive module, which is vectorized to obtain a weighted feature vector. This vector is fed into a feedforward network containing more global information. This operation is then repeated several times on each vector, which takes two linear transformation layers and a ReLU activation function layer. After the end of each sublayer, the data and feature are normalized by the Norm module to ensure the stability of the network's computational gradients.

## Loss function

Object detection mainly contains two types of loss functions, classification, and regression, and the classification loss mostly uses the generic cross-entropy loss. The regression loss has many forms of improvements, which can be applied with the further research progresses. When the model is trained, the loss values are calculated to predict the change between the bounding box and the ground truth box. The SPN-TS discards the original loss function and innovatively introduced new classification and regression functions. The final loss function is as follows,

$$L = L_{cls} + L_{reg} \tag{7}$$

where $L_{cls}$ and $L_{reg}$ are losses of classification and regression, respectively. The classification loss is defined as follows Focal Loss [30] to solve the problem of sample imbalance.

$$L_{cls} = \text{Focal Loss} = \begin{cases} -\alpha(1 - y')^\beta \log y' & , \ y = 1 \\ -(1 - \alpha)y'^\beta \log(1 - y') , \ y = 0 \end{cases} \tag{8}$$

The method adopted CIoU loss function, which to solve the problem of inconsistency between the metric and the border regression on logo detection, and the calculation method is

shown in equation,

$$L_{reg} = 1 - IoU + R_{CIoU}(B_{pd}, B_{gt}) \tag{9}$$

$$L_{reg} = 1 - \frac{B_{pd} \cap B_{gt}}{B_{pd} \cup B_{gt}} + \frac{\varphi^2(b, b_{gt})}{c^2} + \alpha \frac{4}{\pi^2}(arc\tan\frac{w^{gt}}{h^{gt}} - arc\tan\frac{w}{h})^2 \tag{10}$$

Due to the high performance of the proposed SPN-TS method in breast cancer object detection, we strongly recommend it for the diagnosis of multiple diseases, such as lung cancer and brain cancer. Moreover, the proposed method can be easily incorporated into healthcare systems for reliable diagnosis of multiple diseases due to its reproducibility.

## Results

We performed extensive experimental evaluations on the CBIS-DDSM dataset. The experimental design was compared with various state-of-the-art detection methods and medical detection methods to demonstrate the effectiveness of the proposed SPN-TS network. We further performed ablation studies and qualitative analysis to demonstrate the effectiveness of our method in detecting small objects.

### Dataset

CBIS-DDSM is a publicly available breast photographic dataset. The earlier dataset was manually screened by experienced physicians. In this experiment, 2424 complete photographs of benign and malignant breast masses were selected from this data as experimental data. It is an unbalanced dataset containing 1629 benign and 795 malignant tumors. At the same time, high quality and clean detection frame labels are provided manually for the object detection task. The proportion of training verification and test in the experiment was set as 70%, 20% and 10%. Before the experiment, the lossless JPG image was converted to PNG format using the calibration feature of the DDSM website (http://www.eng.usf.edu/cvprg/Mammography/Database.html). The image size was set to 224 × 224, and the pixel values were readjust to be within range from 0 to 255 pixels. A sample image of a mammogram is shown in Fig 4, containing the tumor of varying sizes and different resolutions.

### Experimental setup

The size of the mammogram image input to the network is 224 × 224, and the ReLU function was set as a nonlinear activation function. When these detectors were trained, the initial learning rate is set to 0.001. The learning rate in MMdetection is calculated using the principle of linear scaling [35] to obtain the learning rate of the training model. The learning rate is adjusted downward to 0.0001 when the number of iterations is 10000, to further converge the function loss value. The batch size is 64 and the momentum factor is set to 0.9. Inspired by recent CNN methods [36–38], the experiments implement feature extraction with VGG16 and ResNet-50 as the backbone network. All baseline detectors were re-implemented using the same code base based on the exposed MMDetection (https//github.com/open-mmlab/mmdetection) toolkit for a fair comparison of all methods. All models are trained on the same training set and validated on the validation set. This work was performed on an Nvidia GTX1080Ti GPU and Python was used as the programming language on Ubuntu 14.04 OS.

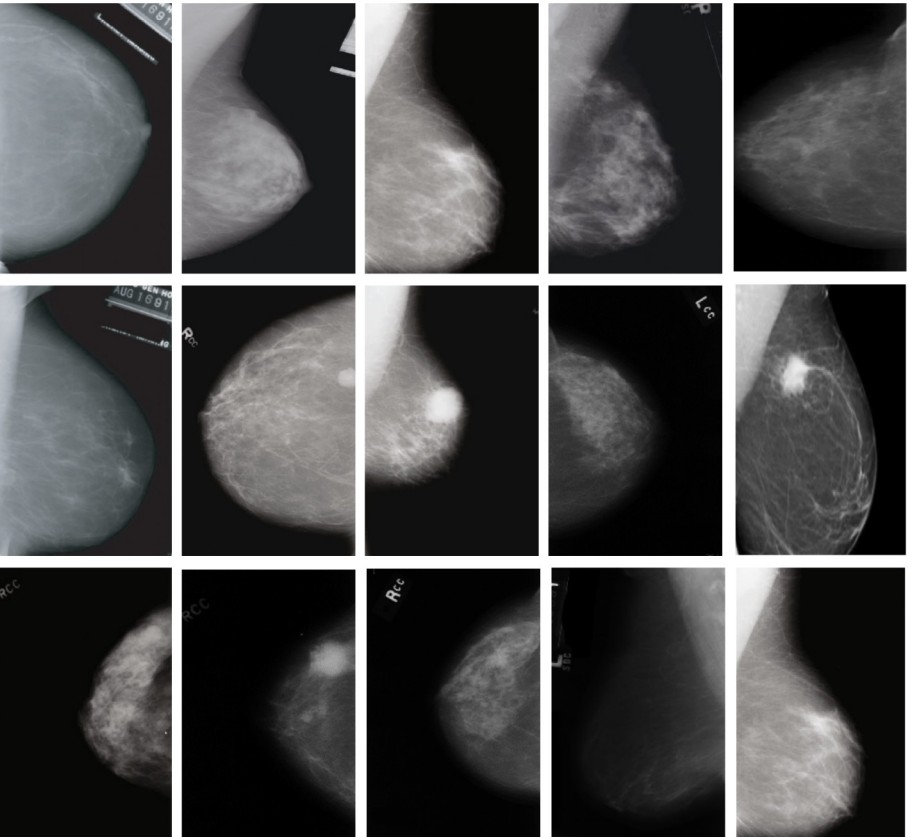

**Fig 4. Some samples of mammogram images.**

## Evaluation metrics

Different evaluation indicators were used to evaluate the performance of the proposed method. Each identified mass requires a specific detector to output its cancer tumor type, which is then compared to a given ground-based reality for one of four options. The first one is a mass classified as benign (True Negative TN), the second is a benign mass classified as malignant (False Positive FP), the third is a malignant mass classified as malignant (True Positive TP) and finally, a malignant mass classified as benign (False Negative FN). The above four indexes constitute the confusion matrix, which is one of the important indexes to evaluate accuracy. In this paper, the above indicators will be used to further calculate several other indicators.

$$Accuracy(ACC) \;=\; \frac{TN + TP}{FP + TN + FN + TP} \tag{11}$$

$$Specificity(FPR) \;=\; \frac{TN}{FP + TN} \tag{12}$$

$$Sensitivity(TPR) \;=\; \frac{TP}{FN + TP} \tag{13}$$

$$Precision \;=\; \frac{TP}{FP + TP} \tag{14}$$

$$F - measure = \frac{2\,\text{Sensitivity} \times \text{Precision}}{\text{Sensitivity} + \text{Precision}} \qquad (15)$$

$$MCC = \frac{\text{TP} \times \text{TN} - \text{FP} \times \text{FN}}{\sqrt{(\text{TP} + \text{FP})(\text{TP} + \text{FN})(\text{TN} + \text{FP})(\text{TN} + \text{FN})}} \qquad (16)$$

Eqs (11)–(16) represent Accuracy, mAP, sensitivity, specificity, precision, F-measure, and Matthews Correlation Coefficient (MCC), respectively. For evaluation, we used the widely used mean accuracy (mAP) to evaluate the network detection results. IoU threshold set to 0.5 mAP, which denotes the precision and recall as the region included in the two-axis mapping. The N is represents mean value. P represents the average accuracy rate of each category, which is used to evaluate the detection results of each category. The higher its value, the better the performance of the model.

## Comparison with detector baselines

**Results.** In order to prove the effectiveness of the proposed model for the detection of cancer tumors, the paper first compares the SPN-TS method with various popular object detection models. Among them, the classical two-stage detection method Faster R-CNN [29] and the single-stage detector SSD [33] and YOLOv3 [32] are included. There are also recently improved proposed FPN [9], mask R-CNN [39] and Cascade R-CNN [31] object detectors for experiments. All of the above detectors use ResNet-50 as a backbone network for feature extraction, which ensures a uniform scale.

Table 1 summarizes the results of CBIS-DDSM in different detection models. Compared with the existing baselines such as Faster RCNN, YOLOV3 and Mask R-CNN, Cascaded R-CNN detectors in all baselines achieved 89.81% mAP and 93.61% ACC, respectively. This detector achieves better results compared with other generic detectors, therefore, Cascaded R-CNN is used as the baseline in this paper. Faster-RCNN, a popular detector, also achieved 83.20% mAP detection performance because there are more small tumor objects and fewer objects in many images in the real world. We then compared the performance of SPN-TS with all baselines, and the mAP metric achieved 92.6%, a 9.4% improvement over Faster-RCNN. As seen in Table 1, it was found that SPN-TS achieved better performance in these models, which achieved 93.26% sensitivity, 95.26% specificity, 96.78% accuracy, and 87.27% MCC value, respectively. It is worth noting that the performance of SPN-TS is improved by 9.4% m AP, 8.66% sensitivity, 10.14% specificity, 15.44% precision, 15.47% accuracy, and 15.07% MCC value, compared with the Faster-RCNN method. The above results demonstrate the significant improvement in detection and classification performance of our

**Table 1. Comparison of the method performance with existing general object detection.**

| Method | mAP | Sensitivity | Specificity | Precision | MCC | ACC |
|---|---|---|---|---|---|---|
| Faster-RCNN [29] | 83.20 | 84.60 | 85.12 | 82.22 | 72.70 | 81.31 |
| SSD [33] | 83.90 | 84.80 | 85.66 | 86.91 | 76.20 | 86.30 |
| YOLOV3 [32] | 85.10 | 86.10 | 87.80 | 89.31 | 79.20 | 89.00 |
| FPN [9] | 88.70 | 90.00 | 91.00 | 90.92 | 81.00 | 90.50 |
| Mask-RCNN [39] | 89.31 | 90.86 | 91.37 | 93.57 | 82.09 | 92.50 |
| Cascade R-CNN [31] | 89.81 | 91.19 | 92.80 | 94.11 | 83.20 | 93.61 |
| SPN-TS (Ours) | **92.60** | **93.26** | **95.26** | **97.66** | **87.27** | **96.78** |

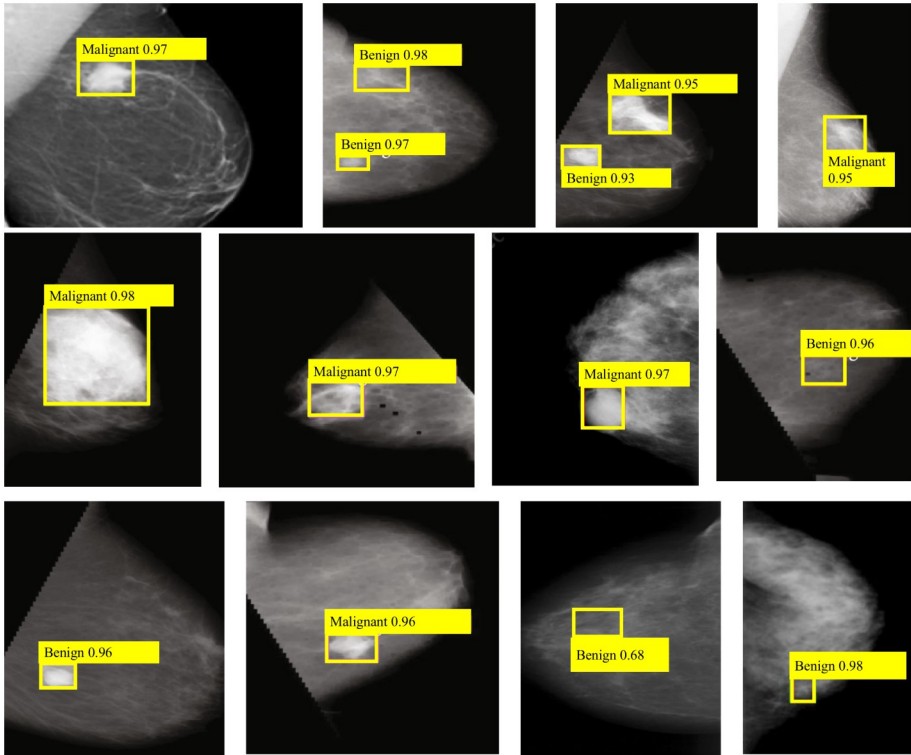

**Fig 5. Results of breast tumor detection and classification using SPN-TS network.**

proposed method, which is beneficial for localization and treatment of irregular and ambiguous tumor regions.

**Analysis.** In addition to the quantitative results of the above experiments, in this section, Fig 5 further presents some detection results of SPN-TS method, including the predicted boundary boxes and classification accuracy. The yellow boxes represent the predicted results. Apparently, the SPN-TS detector can detect larger tumor regions, either benign or malignant, and obtain accurate results, as shown in the first result in the second row of Fig 5. In Fig 5, we show the detection of more small tumor regions and border blurred regions. The second image in the first row with blurred tumor region, the SPN-TS method detects both tumors in the image and results in benign tumors with 95% and 98% accuracy, respectively. The fourth image in the third row possesses a smaller tumor size, and the model can still return 98% of benign tumor detection results, indicating that the method can accurately return smaller tumor regions. The detector also shows high performance for the detection of malignant tumors with blurred edges and irregularities, and is able to regress malignant tumors to identify more accurate regions.

In addition, to more comprehensively analyze the detection effect of SPN-TS method, the Precision-Recall (PR) curve and Receiver Operating Characteristic (ROC) curve were drawn in Fig 6. The PR curve is a curve drawn with Precision as the vertical axis and Recall as the horizontal axis. Therefore, PR is more concerned with the classification of positive samples. The ROC curve uses the FPR values as the horizontal axis and the TPR values as the vertical axis. Therefore, the ability of the model can be accurately judged without the influence of positive and negative sample distribution. As shown in Fig 6, the detection results are better than the original Faster RCNN method in both positive and negative sample processing.

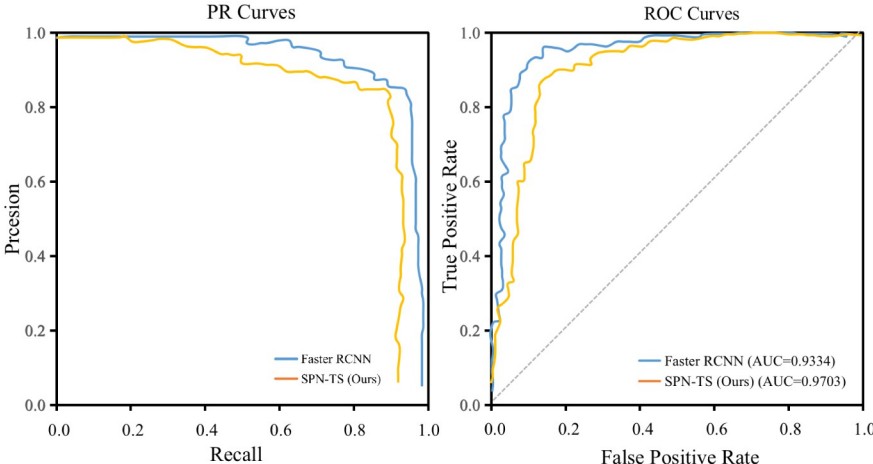

**Fig 6. Comparison of PR and ROC curves between Faster RCNN and SPN-TS.**

## Comparison with medical baselines

**Results.** In addition, the article focuses on the diagnosis of the disease and we will compare the performance of SPN-TS with the better CAD systems available, using the detection of breast cancer as an example. The results generated will also be analyzed in depth. Our choice of the baseline is based on the principle of identification methods that have performed well in recent years in the field of breast tumor detection, mainly containing the classical cancer detection arfLAD [40], RCNN-SVM [40] and DeepCAD [41]. There are also recent BCDROID [26], Cascade R-CNN [31], FMSVM and SC-FU-CS RCNN methods with better detection performance.

As shown in Table 2, it shows the AUC and ACC two indicators for breast tumor detection methods. The two evaluation indicators of BC-DROID method, reached 93.50% and 92.31%, respectively. And the newly proposed SC-FU-CS RCNN method achieved 94.06% accuracy and 94.71% AUC. Our SPN-TS method achieved a 2.72% accuracy improvement over the SC-FU-CS RCNN method to reach the best accuracy of 96.78%. Meanwhile, the AUC reaches 96.78%, which is 2.32% higher than that of the SC-FU-CS RCNN method. In this paper, Table 2 shows that the proposed method achieves the realizes the state-of-art algorithm and solves the problem of identifying multi-scale or ambiguous breast masses.

**Analysis.** In addition to the graphs of detection results given in the previous section, as shown in Fig 7, this paper also shows the tumor visualization results of some test images. The

**Table 2. Comparison of the method performance with existing medical methods.**

| Method | ACC | AUC |
|---|---|---|
| LAD | 78.57 | - |
| RCNN-SVM | 87.20 | 94.00 |
| DeepCAD | 91.00 | 91.00 |
| BC-DROID | 93.50 | 92.31 |
| Cascade R-CNN | 92.76 | 92.72 |
| FMSVM | 91.65 | 96.00 |
| SC-FU-CS RCNN | 94.06 | 94.71 |
| SPN-TS (Ours) | **96.78** | **97.03** |

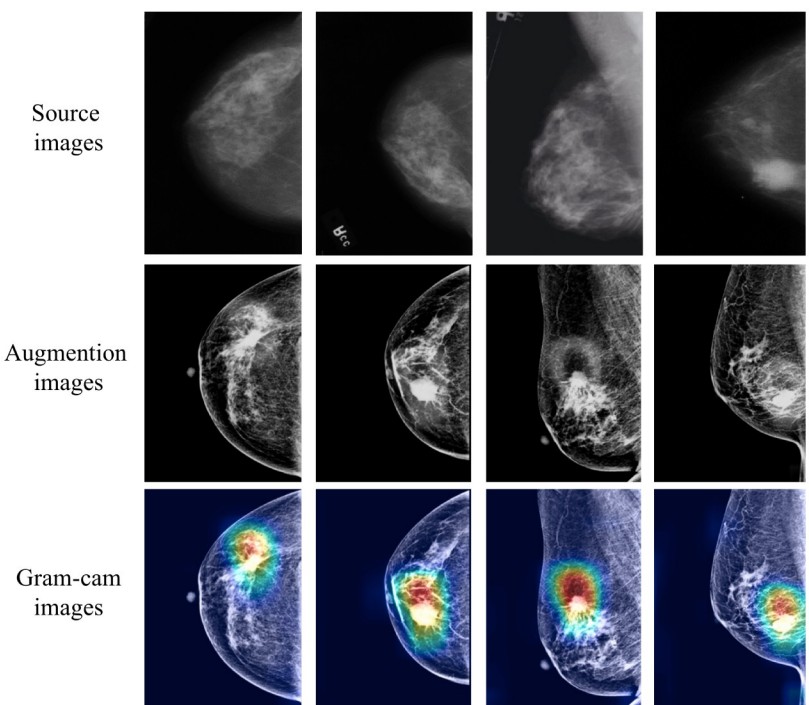

**Fig 7. Gradient-weighted class activation map showing cancerous regions of breast cancer.** The figure contains source data, processed enhanced images, and gram-cam images.

first row is an example of the source image, and the second row shows a simple cut of the breast tissue and sharpening of the tumor region on the original image. The third row of them gives the tumor region highlighted in red, which is displayed more accurately after using the transformer attention module, thus proving the effectiveness of our method. The proposed model effectively detects malignant lesions, which focuses on the interface between breast cancer and the surrounding.

## Ablation study

Table 3 shows a combined analysis of the two modules based on baselines Faster RCNN. First, adding the two modules to the Faster RCNN, the mAP results improved by 5.02% and 4.11%, respectively, which proves the effectiveness of the two modules of semantic feature pyramid network and Transformer Attention. The final mAP result of our method adding two modules is 92.60%, which achieves the best detection result. The mAP, Sensitivity, Specificity, Precision, MCC, and ACC of each index of the ablation experiment were improved by 9.4%, 8.66%, 10.14%, 15.44%, 14.57%, and 15.47%, respectively, compared with the basic baseline. The performance improvement by adding SPN module over TS module is 0.91%, 1.26%, 1.38%,

**Table 3. The detection effects of the two modules based on baselines Faster RCNN.**

| Method | mAP | Sensitivity | Specificity | Precision | MCC | ACC |
|---|---|---|---|---|---|---|
| Faster RCNN | 83.20 | 84.60 | 85.12 | 82.22 | 72.70 | 81.31 |
| Faster RCNN+SPN | 88.22 | 89.94 | 91.35 | 92.82 | 82.56 | 91.00 |
| Faster RCNN+TS | 87.31 | 88.68 | 89.97 | 90.27 | 81.90 | 90.33 |
| SPN-TS (Ours) | **92.60** | **93.26** | **95.26** | **97.66** | **87.27** | **96.78** |

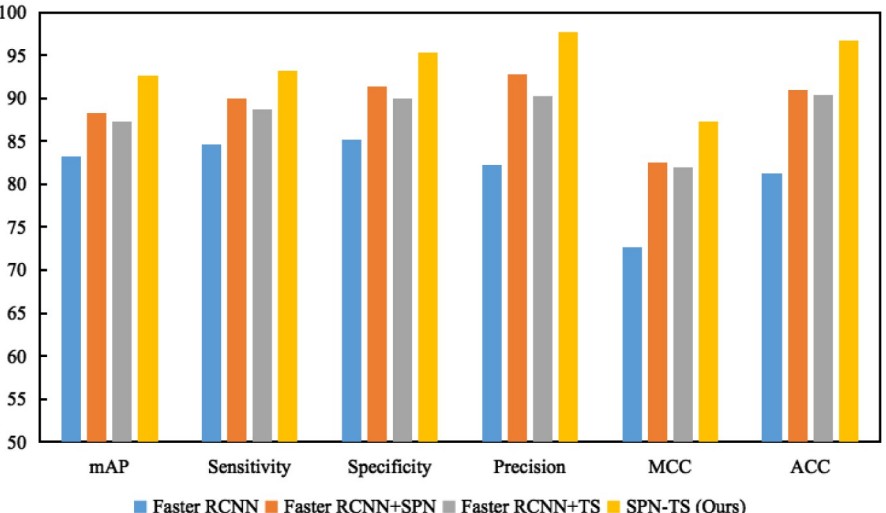

**Fig 8. The histogram results of the ablation experiment showed.**

2.55%, 0.66%, 0.67% for mAP, Sensitivity, Specificity, Precision, MCC, and ACC, respectively. The experimental results indicate that the two designed modules are effective in tumor detection. The results show that our suggested model successfully identifies and classifies breast tumors. In this paper, in addition to presenting a table of ablation experiments, we have further represented the performance increments as bar charts in Fig 8 to see more clearly the degree of effectiveness of each module.

## Discussion

One of the major causes of increased mortality from breast cancer is screening for tumors that are found at an advanced stage, making it difficult to treat effectively. Tumor regions in medical images are usually small in size and fuzzy in boundary. Although the existing research on tumor detection has made a good improvement in feature extraction and target detection algorithm, its performance has not achieved the expected results, especially for the detection of small size tumor. In this paper, we propose an object detection scheme SPN-TS, including semantic feature pyramid network and converter attention module, which specifically solves small tumor targets and lacks context information. An extensive evaluation of state-of-the-art detection methods on the CBIS-DDSM dataset demonstrates the validity of the proposed SPN-TS network.

Compared with existing methods, our SPN-TS method obtains better detection performance, especially in solving small objects and fuzzy boundaries of medical images. We compared the performance of SPN-TS with all baselines, and the mAP metric achieved 92.6%, a 9.4% improvement over Faster-RCNN. Our SPN-TS method achieved a 2.72% accuracy improvement over the SC-FU-CS RCNN method to reach the best accuracy of 96.78%. The final mAP result of our method adding transformer feature extractor and semantic feature pyramid network modules is 92.60%, which achieves the best detection result. The mAP, Sensitivity, Specificity, Precision, MCC, and ACC of each index of the ablation experiment were improved by 9.4%, 8.66%, 10.14%, 15.44%, 14.57%, and 15.47%, respectively, compared with the basic baseline.

However, it can not achieve high detection performance in some cases. The accuracy of small-size tumor detection for medical images still needs to be improved. During the tumor

detection process, we only used medical image data as the input of the model, and did not take into account factors such as the patient's age, gender, and family history, which are parts that need to be improved in the future. In addition, the number of patient samples is also a limitation of this study, and we will collect and label more data from hospitals to support and extend our work in the future. Furthermore, we will use lightweight methods to achieve faster and more accurate performance for tumor detection.

## Conclusion

To solve the tumor of small-size and fuzzy boundaries, the paper proposes a novel detection method by cascading semantic feature pyramid networks and a transformer attention module in medical images. The Semantic FPN obtains semantic features by integrating high-level features into low-level feature maps. The transformer attention model has been used in a variety of object detection tasks and has proven to capture effective key regions. The experiment section demonstrates the effectiveness of the proposed SPN-TS by doing extensive evaluations in CBIS-DDSM dataset, including ablation studies and qualitative analysis. Our algorithm can provide more accurate suggestions for radiologists to diagnose breast cancer and reduce the number of operations for benign breast nodules. The method achieves the best detection performance and effectively solves the difficulties of early tumor detection. In the future, the algorithm is expected to object detection of other diseases. It further provides algorithmic references for the general object detection field. The breadth of the proposed model can be extended to other disease diagnosis and vision-related fields.

## Author Contributions

**Conceptualization:** Mei Lv, Yunfeng Shi.

**Data curation:** Zeyu Han.

**Formal analysis:** Zeyu Han.

**Funding acquisition:** Jingzhen He, Mei Lv.

**Investigation:** Baojun Li, Mei Lv.

**Methodology:** Jing Wang, Yunfeng Shi.

**Project administration:** Baojun Li.

**Resources:** Baojun Li.

**Software:** Baojun Li, Yunfeng Shi.

**Supervision:** Jingzhen He, Yunfeng Shi.

**Validation:** Jing Wang, Zeyu Han.

**Visualization:** Jing Wang.

**Writing – original draft:** Jing Wang.

**Writing – review & editing:** Jingzhen He, Mei Lv, Yunfeng Shi.

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
