## [Decision Letter · Decision Letter 0]

22 Jul 2022

PONE-D-22-17707Cancer Detection for Small-size and Ambiguous Tumors based on Semantic FPN and TransformerPLOS ONE

Dear Dr. Wang,

Thank you for submitting your manuscript to PLOS ONE. After careful consideration, we feel that it has merit but does not fully meet PLOS ONE’s publication criteria as it currently stands. Therefore, we invite you to submit a revised version of the manuscript that addresses the points raised during the review process.

We look forward to receiving your revised manuscript.

Kind regards,

Mohamed Hammad, Ph.D.

Academic Editor

PLOS ONE

Journal Requirements:

A clean copy of the edited manuscript (uploaded as the new *manuscript* file).

NO authors have competing interests. 

6. Please ensure that you refer to Figure 6 in your text as, if accepted, production will need this reference to link the reader to the figure.

Additional Editor Comments:

When updating your manuscript, you should elaborate on your points and clarify with references, examples, data, etc. Also, note that if a reviewer suggested references, you should only add those that are relevant to your work if you feel they strengthen your article.

Reviewers' comments:

Reviewer's Responses to Questions

**Comments to the Author**

1. Is the manuscript technically sound, and do the data support the conclusions?

Reviewer #1: Yes

Reviewer #2: Yes

2. Has the statistical analysis been performed appropriately and rigorously? 

Reviewer #1: Yes

Reviewer #2: Yes

3. Have the authors made all data underlying the findings in their manuscript fully available?

Reviewer #1: Yes

Reviewer #2: Yes

4. Is the manuscript presented in an intelligible fashion and written in standard English?

Reviewer #1: Yes

Reviewer #2: Yes

5. Review Comments to the Author

Reviewer #1: Summary: The authors propose a novel semantic feature pyramid network with a transformer self-attention module, named SPN-TS, for tumor detection, specifically designed to detect small-sized breast tumors with vague boundaries. For the same, authors first create a new FPN feature extraction scheme in the feature extraction stage for small-sized tumor regions, which is followed by the introduction of transformer attention mechanism to capture the features of local tumor boundaries. The technique achieves very good performance while beating all the major state-of-the-art techniques.

My Conclusion: I believe the manuscript is well structured and generally well written. The technique also seems to be sound and achieves very good performance as well. As the authors claim, I believe the technique can be used for variety of image-based prediction tasks. The only issues are that of the grammar and lack of clarity at some places. Most importantly, there is no section for related works, which needs to be introduced. Having said that, I believe the manuscript needs revision, which I classify as a minor one, for acceptance at Plos One, based on the minor comments I provide below.

Comments: My comments are organised and ordered from the start to the end of the manuscript.

1. In Abstract, the authors write, “… we novelty propose… ”. This needs to be corrected to “… we propose a novel…”. This mistake is present throughout the paper and the authors are advised to correct it wherever needed.

2. In Abstract, what are “SPN-TS”, “FPN” and “CBIS-DDSM”? Full forms are needed here.

3. In Introduction, Paragraph 1, Sentence 2, “Breast cancer”, b needs to be small here. Besides, relevant and strong references are needed here as well.

4. In Introduction, Paragraph 1, “As the large number of mammograms performed daily in hospitals, … ” . Grammar check needed here.

5. In Introduction, Paragraph 1, “Firstly, there is a huge difference between cancerous and noncancerous breast tissue … ”. How is this an issue, if there is a “huge” difference between the two? Shouldn't this difference mean that the two are easily differentiable?

6. In Introduction, Paragraph 2, full form of SVM should be provided.

7. In Introduction, Paragraph 2, “General object detection is often applied to automatic detection of cancer images, especially breast masses citeHaq2019Feature, roslidar2020review. Two-stage detectors are a series of R-CNN and RPN modules designed to address the challenges of multi-scale objects citeRen2015, Tsung2017Focal, Cai2018CascadeRCNN, Lin2017FPN. The tyoical one-stage YOLO citeRedmon2016YOLO, Platania2017Automated and SSD citeliu2016ssd detectors are also designed to detect the diversity of object sizes.” There are multiple citation issues which need correcting. Full forms of R-CNN and RPN are needed. When the authors talk about “Two-stage detectors”, this comes out of nowhere and needs some background to stay coherent with the on-going narrative of the paragraph.

8. In Introduction, Paragraph 3, Sentence 1 needs grammar check. If T and A are to be kept capital, perhaps authors could add acronym (TA).

9. In Introduction, Paragraph 3, Sentence 2. Again, what is FPN? Full form and explanation needed. The authors write “… FPN 'first constructs' …”. It is expected that FPN does something more after this phase. This is not clear in the text.

10. In Introduction, Paragraph 3, Sentence 3. Is “general FPN” different from the FPN earlier mentioned? If yes, citation needed here.

11. In Introduction, Paragraph 3. “This will lead to the lack of high-level feature map with sufficient resolution and lacking location information for detecting small objects.” Grammar check required. Perhaps, the authors mean, "This can lead to... ".

12. In Introduction, Paragraph 3. What do the authors mean by feature coupling? How does feature coupling affect the performance?

13. In Introduction, Paragraph 3. “Since it is difficult to extract the information of small tumor objects and the blurring of tumor boundaries in medical images. It is necessary to design an effective method to enrich the high-resolution features and local attention features with semantic information from high-level feature maps.” Grammar check needed. Perhaps the authors meant "... medical images, it is necessary to design...". I think the authors need to put in more background and rationale behind the question of why it is hard to capture small tumor objects and tumor boundaries.

14. In Introduction, Paragraph 4, Sentence 2. Grammar check required.

15. In Introduction, Paragraph 4, Sentence 4. “… we propose a object”. Grammar check needed.

16. Fig 1 needs a lot of explaining. What are Fs and what are Bs. Authors need to provide a brief explanation of how FPN and SPN work in context of the figure itself. As of now, the figure does not explain anything. If it is explained later, the author should mention this and still give a brief overview of the figure.

17. In Introduction, Paragraph 4, last Sentence. Check spellings.

18. Before starting the contribution points, the authors should add a few lines in the previous paragraph starting that the following are their contributions.

19. Contribution 2, why is T capital in "Transformer"? Check this throughout the manuscript.

20. Fig 2, “… transformer attention feature extractor… ”. Comma might be missing here.

21. Section Overview, Paragraph 1, Sentence 3. Grammar check.

22. Section Overview, Paragraph 1, Sentence 7. Grammar check.

23. Section Overview, Paragraph 1. “It helps extract clearer and richer semantic features through the combined contextual information.” How does it work and how is it able to extract clearer and richer semantic features?

24. Section Overview, Paragraph 1, last Sentence. Grammar check.

25. Section Semantic Feature Pyramid Network, Paragraph 1, ROIAlign and RPN need full forms.

26. Section Semantic Feature Pyramid Network, Paragraph 2. “… and is the feature map of Ith stage in the intermediate process through the transverse connection defined as Mi.” ith stage of what? The sentence is not clear.

27. Section Transformer Attention Module, Paragraph 2. What is ResNet-50. Needs full form and brief introduction.

28. Section Results, “the CBIS-DDSM dataset”. Citation required. Web link in the footnote will be good as well.

29. Section Dataset, “and contains 2[U+FF0C]424”. Textual error.

30. Section Dataset, last Sentence, “A sample image of a mammogram is shown in Fig. 4 and contains tumor areas of varying sizes and sharpness.” These are multiple images. Grammar check needed.

31. Section Experimental Setup, Sentence 2. Grammar check needed.

32. Section Experimental Setup, “The learning rate is adjusted downward to

0.0001… ”. By what factor?

33. Section Experimental Setup, what is “VGG16”?

34. Section Experimental Setup, “ImageNet” needs Citation/web link. Brief introduction will also be good.

35. Section Experimental Setup, “… the exposed MMDetection toolkit… ”. What do the authors mean by "exposed"? Citation/web link is also need.

36. Section Comparison with detector baselines, SubSection Results, Paragraph 1, “There are also recently improved proposed FPN /citeLin2017FPN, mask-rCN /citeHe2017Mask and Cascade R-CNN /citeCai2018CascadeRCN object detectors for experiments.” Reference issues.

37. Section Comparison with detector baselines, SubSection Results, Paragraph 2, “Table refobject summarizes… ”. Missing Table reference.

38. Table 1, SSD reference missing.

39. Section Comparison with Medical Baselines, Subsection Results, Paragraph 2, AUC and ACC need full forms.

40. Full fledged Related Works Section needs to be introduced, covering all the significant research conducted in the past pertaining to the manuscript.

41. Dataset needs to be explained in more detail. For example, how many samples are benign and how many malignant, whether there is data imbalance, etc.?

Reviewer #2: Cite the following latest articles published on CNN

1. @article{sahoo2022real,

title={Real-Time Hand Gesture Recognition Using Fine-Tuned Convolutional Neural Network},

author={Sahoo, Jaya Prakash and Prakash, Allam Jaya and P{\\l}awiak, Pawe{\\l} and Samantray, Saunak},

journal={Sensors},

volume={22},

number={3},

pages={706},

year={2022},

publisher={Multidisciplinary Digital Publishing Institute}

}

2. @article{allam2020spec,

title={SpEC: A system for patient specific ECG beat classification using deep residual network},

author={Allam, Jaya Prakash and Samantray, Saunak and Ari, Samit},

journal={Biocybernetics and Biomedical Engineering},

volume={40},

number={4},

pages={1446--1457},

year={2020},

publisher={Elsevier}

}

3. @incollection{allam2022customized,

title={Customized deep learning algorithm for drowsiness detection using single-channel EEG signal},

author={Allam, Jaya Prakash and Samantray, Saunak and Behara, Chinmaya and Kurkute, Ketan Kishor and Sinha, Vikas Kumar},

booktitle={Artificial Intelligence-Based Brain-Computer Interface},

pages={189--201},

year={2022},

publisher={Elsevier}

}

4. @article{venkata2021deep,

title={Deep review of machine learning techniques on detection of drowsiness using EEG signal},

author={Venkata Phanikrishna, B and Jaya Prakash, Allam and Suchismitha, Chinara},

journal={IETE Journal of Research},

pages={1--16},

year={2021},

publisher={Taylor \\& Francis}

}

5. @article{govinda2020review,

title={Review of the Convolution Neural Network Architectures for Deep Learning},

author={Govinda Rao Locharla , Jaya Prakash Allam , Y.V Narayana, Yellapu Anusha},

journal={International Journal of Advanced Science and Technology},

volume={29},

number={4},

pages={2251--2262},

year={2020}

}

6. @article{venkata2021brief,

title={A Brief Review on EEG Signal Pre-processing Techniques for Real-Time Brain-Computer Interface Applications},

author={Venkata Phanikrishna, B and P{\\l}awiak, Pawe{\\l} and Jaya Prakash, Allam},

year={2021},

publisher={TechRxiv}

}

6. PLOS authors have the option to publish the peer review history of their article (what does this mean?). If published, this will include your full peer review and any attached files.

Reviewer #1: No

Reviewer #2: No

---

## [Author Response · Author response to Decision Letter 0]

4 Sep 2022

Response to Reviewers

Dear Reviewers,

First, we appreciate the constructive comments, helpful critiques and favorable assessments from all reviewers. Then, listed below please find our one-by-one response to your comment.

Reviewer #1: 

1. In Abstract, the authors write, “… we novelty propose… ”. This needs to be corrected to “… we propose a novel…”. 

We throughout the paper to correct the same mistake, changing the “we novelty propose…” to “we propose a novel…”

2. In Abstract, what are “SPN-TS”, “FPN” and “CBIS-DDSM”? Full forms are needed here.

The full format of the abbreviations was further added. SPN-TS method means Semantic Pyramid Network with Transformer Self-attention. Feature Pyramid Network is PFN. CBIS-DDSM dataset is the Curated Breast Imaging Subset of the Digital Database for Screening Mammography. 

3. In Introduction, Paragraph 1, Sentence 2, “Breast cancer”, b needs to be small here. Besides, relevant and strong references are needed here as well.

We have revised “Breast cancer” to “breast cancer” in sentence. We have added references in this paragraph that are relevant and powerful for the diagnosis of early breast cancer.

[1] Rajput G, Agrawal S, Biyani KN, Vishvakarma SK. Early breast cancer diagnosis 

using cogent activation function-based deep learning implementation on screened 

mammograms. Int J Imaging Syst Technol. 2022; p. 1101–1118.

[2] Lee Y, Huang C, Shih C, Chang R. Axillary lymph node metastasis status prediction of early-stage breast cancer using convolutional neural networks. Comput Biol Medicine. 2021; p. 104206.

4. In Introduction, Paragraph 1, “As the large number of mammograms performed daily in hospitals, … ” . Grammar check needed here.

The grammar have checked. “With the increasing quantity of mammograms in hospitals, manual reading has become complex and time-consuming for radiologists.”

5. In Introduction, Paragraph 1, “Firstly, there is a huge difference between cancerous and noncancerous breast tissue … ”. How is this an issue, if there is a “huge” difference between the two? Shouldn't this difference mean that the two are easily differentiable?

We have corrected the expression of the first challenge. “There are some differences between cancerous and noncancerous breast tissue on imaging, but the difference in early cancer diagnosis is minimal. ” In general, the lesion area and the tissue area vary widely. However, the difference is too small for early-stage cancer imaging, which makes it difficult to extract tumor region features.

6. In Introduction, Paragraph 2, full form of SVM should be provided.

The full form of SVM is Support Vector Machines. We have provided in Introduction section.

7. In Introduction, Paragraph 2, “General object detection is often applied to automatic detection of cancer images, especially breast masses citeHaq2019Feature, roslidar2020review.”There are multiple citation issues which need correcting. Full forms of RPN are needed. When the authors talk about “Two-stage detectors”, this comes out of nowhere and needs some background.

We correct multiple citation issues. “Object detection algorithms are generally applied to the automatic detection of cancer images, especially breast masses [28, 29]....” The full forms RPN is Region Proposal Network. Most importantly, we have introduced the related work section. “Two-stage detectors”Faster-RCNN generates region proposals through the Region Proposal Network (RPN) module, and then performs fine-grained classification and regression [9, 30–32].

8. In Introduction, Paragraph 3, Sentence 1 needs grammar check. If T and A are to be kept capital, perhaps authors could add acronym (TA).

The sentence grammar have checked. “To solve mentioned above problems, the paper proposes a detection framework for medical imaging, named Semantic Pyramid Network with a Transformer Self-attention (SPN-TS).” We change the initial letter of the method to upper case uniformly.

9. In Introduction, Paragraph 3, Sentence 2. Again, what is FPN? Full form and explanation needed. The authors write “… FPN 'first constructs' …”. It is expected that FPN does something more after this phase. This is not clear in the text.

The Full form of FPN is Feature Pyramid Network. The classical Feature Pyramid Network (FPN) algorithm achieves the extraction of multi-scale features through a top-down multi-level architecture. However, the up-sampling operation used by FPN loses the position information of small objects.

10. In Introduction, Paragraph 3, Sentence 3. Is “general FPN” different from the FPN earlier mentioned? If yes, citation needed here.

Both General PFN and FPN refer to algorithms for target detection of natural images. And the semantic PFN designed in this paper refers to the detection of medical images.

11. In Introduction, Paragraph 3. “This will lead to the lack of high-level feature map with sufficient resolution and lacking location information for detecting small objects.” Grammar check required. Perhaps, the authors mean, "This can lead to... ".

“This can lead to a lack of resolution of features and detection of small objects with position information.”

12. In Introduction, Paragraph 3. What do the authors mean by feature coupling? How does feature coupling affect the performance?

Double-head RCNN[1] disentangle the sibling head into two independent branches for classification and localization. For the classification branch, fully connected layers are employed to extract features to obtain confidence for tumor classification. For regression branches, using convolutional layers to learn representations. Therefore, the feature coupling method achieves the performance of the regression branch under boundary ambiguity by different operations on the two branches. The feature coupling method can effectively improve the performance of small-size detection.

[1] Wu Y, Chen Y, Yuan L, Liu Z, Wang L, Li H, et al. Rethinking Classification and Localization for Object Detection. In: IEEE/CVF Conference on Computer Vision and Pattern Recognition (CVPR); 2020. p. 10183–10192.

13. In Introduction, Paragraph 3. “Since it is difficult to extract the information of small tumor objects and the blurring of tumor boundaries in medical images...” Grammar check needed. Perhaps the authors meant "... medical images, it is necessary to design...". Why it is hard to capture small tumor objects and tumor boundaries?

The Grammar has checked. The traditional Feature Pyramid Network (FPN) was designed for multi-scale feature extraction. For small objects, which occupy fewer pixels in medical images, extracting features becomes particularly difficult. FPN uses top-down and horizontal connection structures leads to low-level feature maps lacking high-level semantic information, which cannot achieve effective detection of small objects. In medical images, malignant tumors have blurred boundaries, which will lead to ineffective discrimination in the detection of regression, so a separate convolution of the regression branch is required to extract features.

14. In Introduction, Paragraph 4, Sentence 2. Grammar check required.

Figure 1(A) shows the traditional FPN by top-down and horizontal connection. The

way does not yield sufficient semantic information, resulting in poor feature extraction from small tumor regions.

15. In Introduction, Paragraph 4, Sentence 4. “… we propose a object”. Grammar check needed.

The paper design a novel FPN feature extraction scheme, which changes the traditional connectivity and improves the semantic information.

16. Fig 1 needs a lot of explaining. What are Fs and what are Bs. Authors need to provide a brief explanation of how FPN and SPN work in context of the figure itself. 

We have mentioned Fig.1 and gave a brief overview of the figure. They are used for the extraction of features. The figure compares the original FPN with our proposed SPN connection. Fi is multi-level features from layer 1 to 5, and Bi is the output of the feature pyramid of the i level. (A) The traditional FPN is by top-down and horizontal connection. (B) Enriched the semantic features through the three steps of lateral connectivity, multiple up-sampling, and feature fusion.

17. In Introduction, Paragraph 4, last Sentence. Check spellings.

Finally, we conducted extensive experimental verification on the CBIS-DDSM, which is the Curated Breast Imaging Subset of the Digital Database for Screening Mammography.

18. Before starting the contribution points, the authors should add a few lines in the previous paragraph starting that the following are their contributions.

The sentence “the contribution of this paper is mainly in the following three aspects. ” was added in contribution. 

19. Contribution 2, why is T capital in "Transformer"? Check this throughout the manuscript.

We thoroughly checked the manuscript and corrected any unnecessary capitalization. “The novel transformer attention mechanism is integrated into the extraction network. The network focuses on tumor region features by attention mechanism and location encoding.”

20. Fig 2, “… transformer attention feature extractor… ”. Comma might be missing here.

The image is input to a feature extractor that incorporates the transformer attention module. This module is used as the first module of the SPN-TS method, called transformer attention feature extractor.

21. Section Overview, Paragraph 1, Sentence 3. Grammar check.

Firstly, this paper designs a novel semantic FPN feature extraction scheme, which changes the traditional cross-layer connectivity way.

22. Section Overview, Paragraph 1, Sentence 7. Grammar check.

The overall architecture of SPN-TS method is shown in Fig 2. It is mainly divided into four parts, transformer attention module, semantic feature pyramid network, region proposal network, prediction of classification and regression module.

23. Section Overview, Paragraph 1. “It helps extract clearer and richer semantic features through the combined contextual information.” How does it work and how is it able to extract clearer and richer semantic features?

The proposed SPN module enriches the semantic features through the three steps of lateral connectivity, multiple up-sampling, and feature fusion. The first operation is a lateral connection. The lateral connections preserve the resolution and semantic information of the current layer while facilitating the feature fusion operations in the subsequent steps. Then, the second operation is lateral connection multiple up-sampling. In the feature fusion step, we integrate feature maps from different layers and different sizes. The semantic FPN network enriches the semantic features by integrating high-level features into low-level feature maps and improving the detection performance of small tumor objects.

24. Section Overview, Paragraph 1, last Sentence. Grammar check.

In addition, the paper decouples the features for classification and regression branches. The module uses fully connected layers and convolutional layers for classification and regression branches, respectively.

25. Section Semantic Feature Pyramid Network, Paragraph 1, ROIAlign and RPN need full forms.

The connected feature maps are input into Region Proposal Network (RPN) and Region of Interest Alignment (ROIAlign) respectively.

26. Section Semantic Feature Pyramid Network, Paragraph 2. “… and is the feature map of Ith stage in the intermediate process through the transverse connection defined as Mi.” ith stage of what? The sentence is not clear.

In Fig. 1, Fi are multi-level features from layer 1 to 5, and is the feature map of Ith (1-5) stage in the intermediate process through the transverse connection defined as Mi.

27. Section Transformer Attention Module, Paragraph 2. What is ResNet-50. Needs full form and brief introduction.

ResNet is a feature extraction network with residual structure. And 50 layers ResNet is abbreviated as ResNet-50.

28. Section Results, “the CBIS-DDSM dataset”. Citation required. Web link in the footnote will be good as well.

Before the experiment, the lossless JPG image was converted to PNG format using the calibration feature of the DDSM website1. In this paper, the footnotes "CBIS-DDSM" dataset was added to the experimental description. The download links are "1:http://www.eng.usf.edu/cvprg/Mammography/Database.html"

29. Section Dataset, “and contains 2[U+FF0C]424”. Textual error.

We revised the Textual error. “In this experiment, 2424 complete photographs of benign and malignant breast masses were selected from this data as experimental data.”

30. Section Dataset, last Sentence, “A sample image of a mammogram is shown in Fig. 4 and contains tumor areas of varying sizes and sharpness.” These are multiple images. Grammar check needed.

The grammar was checked. “Fig. 4 shows some samples of mammogram, which contained the tumor of varying sizes and different resolutions.”

31. Section Experimental Setup, Sentence 2. Grammar check needed.

The grammar was checked. “The ReLU function was set as a nonlinear activation function.

”32. Section Experimental Setup, “The learning rate is adjusted downward to

0.0001… ”. By what factor?

When these detectors were trained, the initial learning rate is set to 0.001. The learning rate in MM detection is calculated using the principle of linear scaling to obtain the learning rate of the training model. The learning rate is adjusted downward to 0.0001 when the number of iterations is 10000, to further converge the function loss value.

33. Section Experimental Setup, what is “VGG16”?

VGGNet is a representative deep network that explores the relationship between the depth of CNN and its performance. VGG16 represents the network with 16 layers.

34. Section Experimental Setup, “ImageNet” needs Citation/web link. Brief introduction will also be good.

ImageNet is currently the largest database for image recognition in the world. The models pre-trained on this dataset have a wide range of applications. The link is “https://image-net.org/”.

35. Section Experimental Setup, “… the exposed MMDetection toolkit… ”. What do the authors mean by "exposed"? Citation/web link is also need.

In this paper, the footnotes " MMDetection" toolkit are added to the experimental description. The download links is "https//github.com/open-mmlab/mmdetection". The “exposed” means “published”. 

36. Section Comparison with detector baselines, SubSection Results, Paragraph 1, “There are also recently improved proposed FPN /citeLin2017FPN, mask-rCN /citeHe2017Mask and Cascade R-CNN /citeCai2018CascadeRCN object detectors for experiments.” Reference issues.

We revised the compilation issues in references. “There are also recently improved proposed FPN [9], mask R-CNN [40] and Cascade R-CNN [32] object detectors for experiments.”

37. Section Comparison with detector baselines, SubSection Results, Paragraph 2, “Table refobject summarizes… ”. Missing Table reference.

We added the missing Table reference. “Table 1 summarizes the results of CBIS-DDSM in different detection models.”

38. Table 1, SSD reference missing.

We corrected the missing SSD reference in Table 1. 

39. Section Comparison with Medical Baselines, Subsection Results, Paragraph 2, AUC and ACC need full forms.

ACC is an abbreviation for Indicator Accuracy. AUC (Area Under Curve) is defined as the area enclosed with the coordinate axis under the ROC curve.

40. Full fledged Related Works Section needs to be introduced, covering all the significant research conducted in the past pertaining to the manuscript.

We have added a complete section of related work, which contains the following four main aspects. First, the current state of research in medical image analysis is summarized. Then, studies examples are given in the early and recent stages of disease detection. Third, we talk about more background of “Two-stage and one-stage detectors”. The process of improvement in small object detection and accuracy is presented. Fourth, recent feature decoupling and transformer object detection algorithms are listed as algorithmic inspirations.

41. Dataset needs to be explained in more detail. For example, how many samples are benign and how many malignant, whether there is data imbalance, etc.?

The earlier dataset was manually screened by experienced physicians. In this experiment, 2424 complete photographs of benign and malignant breast masses were selected from this data as experimental data. It is an unbalanced dataset containing 1629 benign and 795 malignant tumors. The proportion of training verification and test in the experiment was set as 70\\%, 20\\% and 10\\%. 

42. The only issues are that of the grammar and lack of clarity at some places. 

We have throughout the paper and have been advised to correct the grammar.

Reviewer #2: 

1. Cite the following latest articles published on CNN.

By reading the full article, the article has cited the latest references related to deep learning CNNs in the appropriate positions.

[1] Sahoo JP, Prakash AJ, P lawiak P, Samantray S. Real-Time Hand Gesture Recognition Using Fine-Tuned Convolutional Neural Network. Sensors. 2022;22(3):706.

[2] Allam JP, Samantray S, Ari S. SpEC: A system for patient specific ECG beat classification using deep residual network. Biocybernetics and Biomedical Engineering. 2020;40(4):1446–1457.

[3] Allam JP, Samantray S, Behara C, Kurkute KK, Sinha VK. Customized deep learning algorithm for drowsiness detection using single-channel EEG signal. In: Artificial Intelligence-Based Brain-Computer Interface. Elsevier; 2022. p. 189–201.

2. Kindly write the abstract is a concise and succinct manner

In the abstract, we carefully integrated the logic with concise writing and corrected the grammatical errors. 

3. In abstract SPN-TS, CBIS-DDSM full form missed

The full format of the abbreviations was further added. SPN-TS method means Semantic Pyramid Network with Transformer Self-attention. CBIS-DDSM dataset is the Curated Breast Imaging Subset of the Digital Database for Screening Mammography.

4. In Table 1 citation missed

We corrected the citation errors in Table1 and several textual errors in the references as pointed out by the reviewers. 

5. et al. should be in italics

We checked all the citations and italicized all et al. cited into italics.

6. Please avoid the word “we” throughout the script

In the paper, we checked that the word "we" was avoided throughout the text and was replaced by "the paper", "this method", etc.

7. MCC full form in the abstract?

The full format of the abbreviations was further added. MCC value means Matthews Correlation Coefficient.

8. Exact motivation of the work in the introduction missed

In the introduction section, we reorganize the motivation of this work. Cancer detection remains a challenging task due to the interference of diseased tissue, the diversity of mass scales, and the ambiguity of tumor boundaries. Secondly, the varying size of cancer masses is one of the challenges of detection, especially when small ones are difficult to detect. The third challenge is the blurring of tumor boundaries, which may cause visual confusion. Thus, it is necessary to design an effective method to enrich the high-resolution features and local attention features with semantic information from multi-level feature maps.

9.Learning rate of the network?

When these detectors were trained, the initial learning rate is set to 0.001. 

10.Tuning of the network was missed

The learning rate in MM detection is calculated using the principle of linear scaling to obtain the learning rate of the training model. The learning rate is adjusted downward to 0.0001 when the number of iterations is 10000, to further converge the function loss value.

11.Please remove unnecessary capitalizations in the manuscript.

We checked for grammatical and spelling errors throughout the paper and removed unnecessary capitalization. 

12.Please mention important major contributions only

In the introduction section, we added “The contribution of this paper is mainly in the following three aspects.”

(1) The paper proposes an effective network called SPN-TS for cancer detection, containing Semantic FPN and transformer self-attention mechanisms. It addresses the extraction of small tumor objects and the lack of contextual information.

(2) The novel transformer attention mechanism is integrated into the extraction network. The network focuses on tumor region features by attention mechanism and location encoding.

(3) The paper also decouples the classification and regression branches to improve the classification confidence. The experiment conducted an extensive evaluation on CBIS-DDSM dataset to illustrate the effectiveness of the SPN-TS method in detecting small objects.

13.Quality of the images are very poor, and not visible also

We rechecked the figures in the paper for quality enhancement, especially the methodological framework that was redrawn to make it visible. We use “https://pacev2.apexcovantage.com/Upload” link to quality check all the images in this paper.

14.Needs to include ROC curves, and precision-recall curves

To more comprehensively analyze the detection effect of SPN-TS method, the Precision-Recall (PR) curve and Receiver Operating Characteristic (ROC) curve were drawn in this paper. The PR curve is a curve drawn with Precision as the vertical axis and Recall as the horizontal axis. Therefore, PR is more concerned with the classification of positive samples. The ROC curve uses the FPR values as the horizontal axis and the TPR values as the vertical axis. Therefore, the ability of the model can be accurately judged without the influence of positive and negative sample distribution. In the ROC figure, the detection results are better than the original Faster RCNN method in both positive and negative sample processing.

15.Grammatically needs to recheck again.

We have throughout the paper and have been advised to correct the grammar.

---

## [Decision Letter · Decision Letter 1]

12 Sep 2022

Cancer Detection for Small-size and Ambiguous Tumors based on Semantic FPN and Transformer

PONE-D-22-17707R1

Dear Dr. Wang,

We’re pleased to inform you that your manuscript has been judged scientifically suitable for publication and will be formally accepted for publication once it meets all outstanding technical requirements.

Kind regards,

Mohamed Hammad, Ph.D.

Academic Editor

PLOS ONE

Additional Editor Comments (optional):

Reviewers' comments:

Reviewer's Responses to Questions

**Comments to the Author**

1. If the authors have adequately addressed your comments raised in a previous round of review and you feel that this manuscript is now acceptable for publication, you may indicate that here to bypass the “Comments to the Author” section, enter your conflict of interest statement in the “Confidential to Editor” section, and submit your "Accept" recommendation.

Reviewer #1: All comments have been addressed

Reviewer #2: All comments have been addressed

2. Is the manuscript technically sound, and do the data support the conclusions?

Reviewer #1: Yes

Reviewer #2: Yes

3. Has the statistical analysis been performed appropriately and rigorously? 

Reviewer #1: Yes

Reviewer #2: Yes

4. Have the authors made all data underlying the findings in their manuscript fully available?

Reviewer #1: Yes

Reviewer #2: Yes

5. Is the manuscript presented in an intelligible fashion and written in standard English?

Reviewer #1: Yes

Reviewer #2: Yes

6. Review Comments to the Author

Reviewer #1: The authors have addressed all my comments. I have no further comments and I consider the manuscript fit for acceptance and publication.

Reviewer #2: The authors are incorporated all the suggested comments. The quality of the script is now up to the mark.

Please include the following recently published script before the publication:

1. Hammad, M., Chelloug, S.A., Alkanhel, R., Prakash, A.J., Muthanna, A., Elgendy, I.A. and Pławiak, P., 2022. Automated Detection of Myocardial Infarction and Heart Conduction Disorders Based on Feature Selection and a Deep Learning Model. Sensors, 22(17), p.6503.

7. PLOS authors have the option to publish the peer review history of their article (what does this mean?). If published, this will include your full peer review and any attached files.

Reviewer #1: No

Reviewer #2: **Yes: **Allam Jaya Prakash

---

## [Editor Report · Acceptance letter]

29 Sep 2022

PONE-D-22-17707R1 

Cancer Detection for Small-size and Ambiguous Tumors based on Semantic FPN and Transformer 

Dear Dr. Wang:

I'm pleased to inform you that your manuscript has been deemed suitable for publication in PLOS ONE. Congratulations! Your manuscript is now with our production department. 

Kind regards, 

on behalf of

Dr. Mohamed Hammad 

Academic Editor

PLOS ONE